# DiffTell: A Comprehensive Dataset for Image Difference Captioning

## Abstract

The image Difference Captioning (IDC) task is to describe the distinctions between two images. However, existing datasets do not offer comprehensive coverage across all image-difference categories. In this work, we introduce a more extensive dataset, *DiffTell*, which encompasses various types of differences between images, including global image alterations, object-level changes, and text manipulations. *DiffTell* includes both newly collected data and filtered data used in previous studies. Additionally, to scale up the data collection without prohibitive human labor costs, we explore the possibility of automatically filtering for quality control. We prove that both traditional methods and recent multimodal large language models (MLLMs) show improved performance on the IDC task after training on the *DiffTell* dataset. We conducted extensive ablation studies to provide a thorough analysis of the performance gain from *DiffTell*. Experiments show *DiffTell* significantly enhances the availability of resources for IDC research, offering a more comprehensive foundation and benchmark for future investigations.

## 1 Introduction

Given the great progress in image generation (Ramesh et al., 2021; Rombach et al., 2022a), disseminating AI-modified fake images can lead to widespread misinformation, erosion of public trust, and manipulation of public opinion on critical issues. Emerging open standards, such as C2PA (Coalition for Content Provenance and Authenticity, 2023), outline provenance frameworks that utilize perceptual hashing techniques to link images found in the public domain with a federated database of original content (Black et al., 2021; Pizzi et al., 2022). Upon retrieving the source image, image difference captioning (IDC) models can describe the discrepancies between the circulated image and its original, enabling individuals to make more informed and nuanced trust assessments. IDC has been researched with various algorithms (Tan et al., 2019; Qiu et al., 2021; Tu et al., 2021; Guo et al., 2022b; Yao et al., 2022b; Tu et al., 2023e;a). However, the image domain and the difference types of the current IDC dataset are either limited or small-scaled, as summarized in Table 1. This makes the generalization ability of the current model unsatisfactory; thus, a comprehensive IDC dataset on a large scale is needed.

The IDC dataset consists of the data triplet, including one image pair (the original and the manipulated) and one language caption describing the difference between them. The formal definition is given in Section 3.1. As shown in Table 1, existing datasets focus either on domain-specific images, such as Spot-the-diff (Jhamtani & Berg-Kirkpatrick, 2018a) with frames of the surveillance videos, or 3D-rendered scenes with limited objects and change types (color, texture, add, drop, remove) in CLEVR (Park et al., 2019b). Even though image editing request (IER) has various types of editing on the real natural images, it is limited in volume ($\sim$ 4K) since manual human editing is costly and time-consuming, making it harder to scale up (Tan et al., 2019). Given the development of generative AI and image editing technologies, language-guided AI-manipulated image data have been created with data triplet: before-edited image, after-edited image, language editing request. InstructPix2Pix (Brooks et al., 2023) leverages GPT-3 (Brown et al., 2020) to scale up possible editing commands and resort to prompt2prompt (Hertz et al., 2022) for automatic editing. However, we find that it includes a high error rate, which is over 60%. MagicBrush (Zhang et al., 2023) provides 10K manually annotated real image editing triplets with careful quality control but only contains local edits. It has showcased the importance of high-quality data for language-guided image editing.

Therefore, we identify a need for an IDC dataset that is varied in manipulation types and maintains high quality at a large scale.

To better support research in image difference captioning, we introduce the *DiffTell* dataset, specifically created to encompass a broader range of editing types, including both real and synthesized image pairs, while maintaining careful quality control. We include four categories of image difference: background change, local object change, text manipulation, and image style change from various data sources. Examples of the *DiffTell* dataset are illustrated in Fig. 1. We first include two accessible language-guided image editing datasets InstructPix2Pix (Brooks et al., 2023) and MagicBrush (Zhang et al., 2023). We manually filtered out the noisy, low-quality data in InstructPix2Pix. As text manipulation is critical in creating fake news, we enriched the text addition and removal data by inpainting the text in MARIO-10M images (Chen et al., 2023a). In addition, we extended the object addition and removal by inpainting the COCO (Lin et al., 2014) dataset. All AI-generated editing outcomes have passed the quality filtering process. Moreover, since the labor cost of manual quality filtering could be expensive when scaled up, we further learn an automatic data filtering model to reduce the cost and observed the benefit of such an auto filtering process according to model captioning performance.

Multimodal large language model (MLLMs) have become increasingly popular in the research community due to their strong general-purpose capability. By linking large language models (LLMs) with visual conditioning (Liu et al., 2023e; Zhu et al., 2023), MLLMs have shown impressive results in natural instruction-following and visual reasoning capabilities. Meanwhile, the *DiffTell* dataset can serve as a visual instruct finetuning (Liu et al., 2023e) step upon the multiple MLLM models. We demonstrate the general improvement of IDC performance using the *DiffTell* dataset on various baselines, indicating its value and benefits. In summary, our contributions are

- Proposing the *DiffTell* dataset that includes various kinds of changes with high-quality samples on a larger scale than previous datasets;
- Proving that *DiffTell* can boost the IDC on various baselines on both IER and PSBattle datasets;
- A detailed analysis of how the *DiffTell* dataset enhances IDC in different editing categories;
- Probing the model-based data filtering given the fixed amount of human-filtered data, allowing potential data scale-up.

Table 1: The comparison involves *DiffTell* and currently available datasets designed for the image difference captioning (IDC) task. "Real" and "Syn." signify the presence of real and synthetic images in the datasets, respectively. "Human Anno." indicates whether the dataset is filtered with human annotations. The term "comprehensive" category denotes that the dataset can encompass all the categories outlined in Section 3.2. A more detailed existing dataset description is given in Section A.

| Dataset | Size | Real | Syn. | Human Anno. | Categories | Domain |
|---|---|---|---|---|---|---|
| CLEVR-Change | 70K | ✗ | ✓ | ✗ | Local object | primitive 3D shapes |
| Spot-the-Diff | 13K | ✓ | ✗ | ✓ | Local object | top-down street view |
| IER | 4K | ✓ | ✗ | ✓ | Comprehensive | varied natural images |
| PSBattl | 100 | ✓ | ✗ | ✓ | Comprehensive | varied natural images |
| *DiffTell* (Ours) | 70K | ✓ | ✓ | ✓ | Comprehensive | varied natural images & genAI |

## 2 RELATED WORK

### 2.1 MULTIMODAL LARGE LANGUAGE MODELS

With the development of visual encoder and its combination to large language models (LLMs), multimodal large language models (MLLMs) (Liu et al., 2024; 2023c;d; Zhu et al., 2023) show promising capability to understand images, accept text inputs, and generate natural-language responses. Increasing the model capacity and dataset size can generally improve the capability of MLLMs (Zhang et al., 2022; Bai et al., 2023; Chen et al., 2023c). Visual encoders (Radford et al., 2021; Li et al., 2022; 2023) are applied to encode visual information into visual tokens, providing input for the LLMs. Other strategies like expanding the instruction-tuning dataset (Liu et al., 2023a) and increasing the visual resolution (Wang et al., 2023; Bai et al., 2023; Liu et al., 2023b) can also improve the performance of the MLLMs. Recently, MLLMs have been used to understand fine-grained images, such as in local

region understanding (Chen et al., 2023b; Liu et al., 2023f). Image difference captioning is closely related to fine-grained image understanding with multiple-image input.

## 2.2 IMAGE DIFFERENCE CAPTIONING

As mentioned above, MLLMs are used to understand the local region. Image difference captioning (IDC) is more challenging because the model needs to not only understand each image correctly but also capture and identify the difference between two images correctly and express it precisely in language. In IDC, the caption aims to describe the differences between the images while ignoring their commonalities. The first work on IDC, Spot-the-Diff (Jhamtani & Berg-Kirkpatrick, 2018b), categorizes different types of changes and uses an LSTM-based network to model them. DUDA (Park et al., 2019a) improves the robustness against slight global changes by analyzing image differences at a CNN semantic level instead. Viewpoint invariant encoders have been proposed in M-VAM (Shi et al., 2020b), VACC (Kim et al., 2021), and VARD (Tu et al., 2023c) to mitigate potential viewpoint differences, while (Sun et al., 2022) uses bidirectional encoding to improve change localization and NCT (Tu et al., 2023d) aggregates neighboring features with a transformer. IDC-PCL (Yao et al., 2022a) and CLIP4IDC (Guo et al., 2022a) adopt BERT-like training strategies to model the difference-captioning language. SCORER (Tu et al., 2023f) applies a self-supervised cross-view representation reconstruction technique for difference captioning. Recently, with the advancement of MLLMs, more datasets have integrated the existing IDC dataset to train powerful MLLMs with diverse capabilities. For instance, LLaVA-OneVision (Li et al., 2024) includes the CLEVR dataset, while Mantis-Instruct (Jiang et al., 2024) incorporates the Spot-the-Diff dataset.

## 2.3 IMAGE EDITING

One of the biggest challenges in IDC is the shortage of high-quality, comprehensive datasets of paired images. The development of diffusion model (Ho et al., 2020) significantly improves the quality and controllability of the generated images. By controlling the cross-attention, diffusion models can transform the image globally (Rombach et al., 2022a; Saharia et al., 2022). Local editing depends on the fine-grained predicted or user-provided mask, such as inpainting (Lugmayr et al., 2022; Nichol et al., 2021; Avrahami et al., 2022). Different from the image transformation and local editing, the input of the instruction-guided image editing is in the command format rather than the detailed description and mask (Brooks et al., 2023). *DiffTell* significantly benefits from the progress in image generation models (Rombach et al., 2022b), especially the local editing model, leveraging their capabilities to enhance the quality and diversity of the dataset.

## 3 PROBLEM FORMULATION AND DATASET CONSTRUCTION

### 3.1 PROBLEM DEFINITION

For IDC problem, when presented with two similar images, denoted as $I_1$ and $I_2$, our objective is to employ a vision-language (VL) model, $f_\theta$, to articulate the distinctions between $I_1$ and $I_2$ in natural language. This can be represented as: $T_{I_1,I_2} = f_\theta(I_1, I_2)$, where $T_{I_1,I_2}$ represents the descriptive caption text provided by the model regarding the dissimilarities between the images, and $\theta$ signifies the model parameters within the VL model. The elements $I_1$, $I_2$, and $T_{I_1,I_2}$ collectively form the constituents of each sample within the IDC dataset.

### 3.2 IDC CATEGORIES

Considering that our main motivation is to alleviate the misinformation and spreading of doctored images, we focus on the image pairs created by manipulation or editing and exclude the pairs without any correlation or cannot be easily obtained by human/AI editing. To further concretize the research problem, we categorize four image difference types as *background change, local object modification, style change* and *text manipulation*. **Background change** is alterations related to the background, such as removing, adding, or changing the background of an image. **Text manipulation** involves addition, removal, or modification of text within the original image. **Local object change** is about object re-colorization, appearance editing, object removal, insertion, or translation. **Style change** is the artistic style change, such as realistic photo to painting, and photo-realistic style change, such

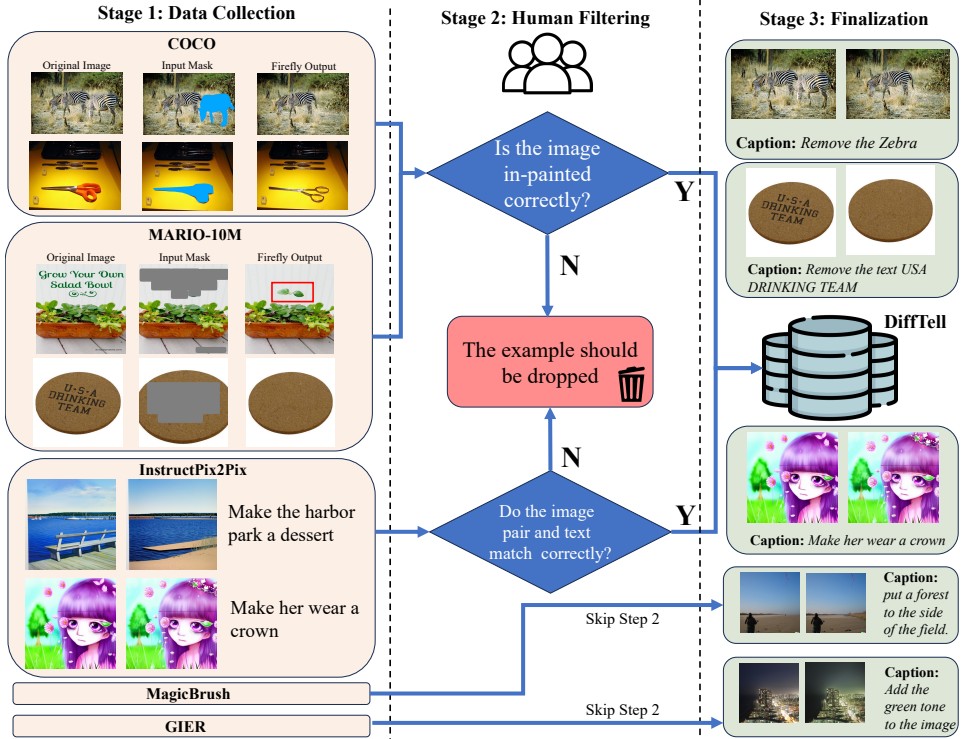

Figure 1: The data collection pipeline involves two steps. Initially, data is gathered from COCO, MARIO-10M, InstructPix2Pix, MagicBrush, and GIER. For COCO and MARIO-10M, an in-painting process is applied to the images with the help of masks, and the labeling team subsequently filters out unsuccessful cases (Step 2). The three images are the original image, the input mask and the output from Firefly Generative Fill from the left to the right. In the second (lower) COCO example, where the scissors remain unaltered, the labeling team excludes this case from the dataset. Similarly, for the first (upper) MARIO-10M example, although the text in green is removed, the generation model introduces an additional element outlined in the red box, leading to the exclusion of this example as well. In the case of InstructPix2Pix, the labeling team verifies the alignment between image pairs and language instructions. Instances with unsuccessful modification (e.g., the dessert modification in the top example) are removed from the dataset. For the MagicBrush and GIER datasets, Step 2 is skipped as they have already undergone manual filtering. The final stage involves compiling the filtered data, resulting in the creation of the DiffTell dataset.

as adjusting the brightness or tone. Existing datasets such as IER mainly include the first three categories but lack text manipulation. However, text manipulation is crucial in our scope since some text changes can flip the message of an image, leading to fake news and forged messages. For example, the message of a smiling face image can be changed from happiness to sarcasm by adding the sentence "absolutely thrilled to be overworked and underpaid." Therefore, we put additional effort into text manipulation data collection. The detailed elaboration of each difference category is as follows.

### 3.3  DATASET COLLECTION PIPELINE

Based on the definition in Section 3.1, the triplet $(I_1, I_2, T_{I_1,I_2})$ reflecting the four categories given above is the fundamental element to build an image difference captioning (IDC) dataset. As the mirrored task of IDC, the instruction-guided image editing dataset is considered, which provides $(I_1, I_2, T_{I_1,I_2})$ exactly. We select InstructPix2Pix (Brooks et al., 2023), GIER (Shi et al., 2020a), and MagicBrush (Zhang et al., 2023) as the subset of our dataset due to the editing types, dataset sizes/qualities. The difference categories of those three datasets are given in Table 2.

Table 2: Summary of the source datasets from which we derived our dataset. "Syn. Image" indicates whether the image domain contains synthetic images, while the "F. rate" denotes the ratio of images retained after manual filtering by our labeling team if needed, which is equal to (100% - Rejection Rate).

| Datasets | Syn. Image | F. Rate (%) | Image Difference Categories | Dataset Size |
|---|---|---|---|---|
| InstructPix2Pix | ✓ | 35.13 | Background, Image style, Local object | 17,592 |
| GIER | ✗ | 100.00 | Background, Local object, Image style | 6,179 |
| MagicBrush | ✗ | 100.00 | Local object, Text | 8,807 |
| MARIO-10M | ✗ | 26.86 | Text | 30,903 |
| COCO | ✗ | 43.87 | Local object | 12,886 |
| *DiffTell* | ✓ | | Comprehensive | 67,589 |

Most existing vision datasets only provide $I_1$ and its corresponding annotations, like the object segmentation mask or the object's name. Empowering the generative model (Rombach et al., 2021; Yang et al., 2023), we can remove an object from the image to generate $I_2$ although a quality check step is necessary due to the limitation of the generative model. The difference caption $T_{I_1,I_2}$ can be generated based on the editing operation from the generative model. For datasets only providing $I_1$, such as COCO and MARIO-10M, we mainly focus on object change and text manipulation. For the generation of $I_2$, we apply the inpainting model Firefly Generative Fill[1] and the details of how to generate images are given in Appendix G. $T_{I_1,I_2}$ is based on the template "Add <Text> / <Object>" or "Remove <Text> / <Object>" depends on the order of $I_1$ and $I_2$, which is determined by a random number generator whose probability is 0.5. For the datasets providing $I_1$, $I_2$ and $T_{I_1,I_2}$ without manually filtering like InstructPix2Pix, we ask the labeling team to filter them. We provide the details of each subset and annotation details below.

**InstructPix2Pix** (Brooks et al., 2023) provides $I_1$, $I_2$ and $T_{I_1,I_2}$, where $(I_1, I_2)$ are generated by StableDiffusion (Rombach et al., 2022a) in combination with Prompt-to-Prompt, and $T_{I_1,I_2}$ is produced by a finetuned GPT-3 (Brown et al., 2020). It is a large (450K+) dataset with various image-difference categories thanks to the automated process. However, the automated process occasionally mismatches the image pair and its corresponding instruction. We present such a noisy sample in Fig. 1. The instruction "Make the harbor park a dessert" does not describe the difference between the image pair. To mitigate this, our labeling team meticulously reviews a subset to retain clear and accurate samples. After reviewing 50,012 selected triplets from the InstructPix2Pix dataset, we obtain 17,592 image pairs covering background, style, and local object change.

**GIER** (Shi et al., 2020a) also provides the $(I_1, I_2, T_{I_1,I_2})$ triplet, presenting 6,179 image pairs. IER and GIER are both from the same source and complementary to each other. More specifically, they are both from the human Photoshop-edited images based on the language editing instructions. GIER is mostly characterized by its global tone and lighting editing. We employ these pairs along with expert annotations as $I_1$, $I_2$, and $T_{I_1,I_2}$ respectively, while standardizing the language style by removing unnecessary politeness indicators like "Please."

**MagicBrush** (Zhang et al., 2023) constitutes a high-quality dataset for multi-turn image editing, meticulously curated through manual filtering, providing $(I_1, I_2, T_{I_1,I_2})$ triplets in high quality, which can be used directly in IDC task. To adapt this multi-turn editing to fit our framework, we segmented it into several single-turn edits and randomized their order. As a result, we incorporate 8,807 image pairs from Magicbrush into *DiffTell*.

**MARIO-10M** (Chen et al., 2023a): Text manipulation data is gathered based on MARIO-10M, a dataset offering rough segmentation masks and optical character recognition (OCR) results for text within images. The dataset only provides $I_1$, and we use FireFly Generative Fill to remove the masked text from the images to generate $I_2$ with the input of $I_1$ and its corresponding mask. We apply mask dilation, enlarging the original mask by 5 pixels to make the region of interest (ROI) covered by the mask as much as possible. Our labeling team carefully verifies the resulting images to ensure that the text is fully removed and there is no additional element added in $I_2$, leading to the retention of 30,903 image pairs out of 115,059 in our dataset. For filtered image pairs $(I_1, I_2)$,

---

[1]As a type of artificial intelligence that can translate text and other inputs into extraordinary results, Firefly Generative Fill model can generate the image according to the image or text input and be accessed at `https://firefly.adobe.com`.

the language templates $T_{I_1, I_2}$ we use are "add text" or "remove text," depending on the order of the image pair. We also add the OCR results to the caption, with examples given in Fig. 1.

**COCO** (Lin et al., 2014): Similar to MARIO-10M dataset, COCO dataset only provides $I_1$ and we need to generate $I_2$ and $T_{I_1, I_2}$. We initially generated masks for each instance from the annotations in the training set. Different from MARIO-10M, the mask cannot be used directly because some of the object masks are tiny, while some occupy almost the whole image, although the object is the same. To ensure proper object sizes, a mask filtering technique is applied, selecting objects within a specific size range based on the distribution of mask sizes within each class. For each class, we select the images with the masks whose area is 50%-75% of the largest area to ensure that the change within the image pairs is obvious and meaningful while not occupying the full image. This process results in a selection of 128,969 images from an initial pool of 860,001. Similar to the MARIO-10M approach, mask dilation is applied in case of potential detail loss in polygon masks. Objects are in-painted using FireFly Generative Fill, and the resulting images are scrutinized by our labeling team, resulting in a final selection of 12,886 image pairs out of 29,374 for our dataset. After getting the image pairs with and without the object from inpainting, we follow the language template in MARIO-10M, which is "add <object>" or "remove <object>" as shown in Fig. 1. The COCO subset in *DiffTell* focuses on local object change.

**Quality Check Statistics** We use LabelBox[2] as our crowdsourcing platform. Each sample added to *DiffTell* is initially labeled by an annotator and then reviewed by a high-performing annotator selected by us. To identify high-performing annotators, we have each annotator label 500 images to assess their understanding of the task, and we manually evaluate their accuracy. The top 30% of annotators are selected as high-performing and assist with the review process on a larger scale. On average, the labeling time is 56.73 seconds, while the reviewing time averages 72.44 seconds.

**Rationality of Data Construction with Generative Model** Considering the circulated deceptive doctored images are usually edited by humans or AI, we also create the image pair with human or AI manipulation. InstructPix2Pix, MagicBrush, Mario-10M, and COCO are AI-edited, and GIER is human Photoshopped. And we can control the type of difference in the dataset based on the editing we applied, allowing future balancing and debias of various IDC categories.

### 3.4 DATASET ANALYSIS

Following the dataset collection, we conduct a statistical analysis of the *DiffTell* dataset based on the four categories in Section 3.2. The contribution to each editing category within each subset of *DiffTell* is presented in Fig. 3b. Background and image style changes are from GIER and InstructPix2Pix. MARIO-10M is for text manipulation. Local object change is from all the subsets except MARIO-10M. Over 72.9% images' resolution is $512 \times 512$. The largest image is $1024 \times 1024$, which is over 10%. The ratio of the images in other resolution is less than 1.5%. The average length of the difference description is 9.72 words. The longest description is 66 words, while the shortest is 3 words. The most descriptions contain 9 words. The description length distribution are given in Figure 2. We attach more dataset illustration and how the labeling team works to filter the data to the Appendix G.

## 4 EXPERIMENTS

### 4.1 EXPERIMENT SETUP

**Benchmark Datasets and Evaluation Metrics** We conduct experiments on the IER dataset (Tan et al., 2019) and the PSBattle dataset (Black et al., 2024), which encompass a wide range of image editing differences. The PSBattle dataset is sourced from the PSRequest channel on Reddit[3], comprising 100 pairs of images, each associated with at least three captions depicting image modifications (Black et al., 2024). Note that we exclude CLEVR (Park et al., 2019b) and Spot-the-difference (Jhamtani & Berg-Kirkpatrick, 2018a) from our evaluation because they only focus on a single image domain (simple geometry and surveillance camera), and their image pairs are not created by human/AI edit, deviating from our motivation of building a dataset with various image difference types to

---

[2]https://labelbox.com
[3]https://www.reddit.com/r/photoshopbattles/

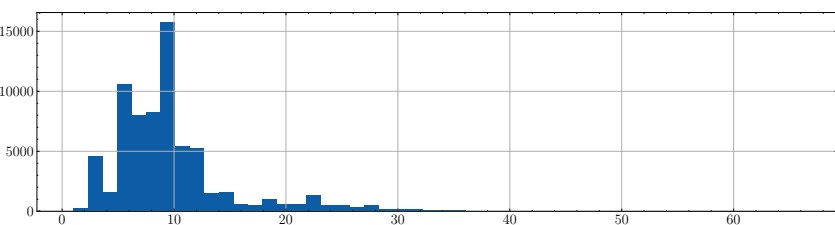

Figure 2: The difference description length distribution in *DiffTell*

avoid deceptive image doctoring. In the case of IER, we evaluate performance on the testing set by comparing models trained exclusively on the IER training set and those trained on a combination of the IER training set and the *DiffTell* dataset. There exists overlap between GIER and IER datasets and we drop the samples in GIER which also exist in the IER testing set. For the PSBattle dataset, we adopt it as an out-of-domain dataset to test the zero-shot capability of our model. Aligned with prior captioning research, we employ BLEU@4 (Papineni et al., 2002) (B@4), METEOR (Banerjee & Lavie, 2005) (M), CIDEr (Vedantam et al., 2015) (C), and ROUGE-L (Lin, 2004) (R-L) as the evaluation metrics.

**Baselines and Implementation Details**   We implement several baseline methods for IDC to comprehensively illustrate the benefits of the *DiffTell* dataset, including both IDC-specific and MLLM methods. For IDC-specific methods, we use CLIP4IDC (Guo et al., 2022b). For MLLM methods, we report OpenFlamingo-3B (Awadalla et al., 2023), Fuyu-8B (Bavishi et al., 2023) and Llave-interleave-8B (Liu et al., 2023c) here. We follow the instruction tuning methods to train the MLLMs. Without further clarification, the prompt we use across all the experiments is "What is the difference between two images?". We also try the diverse instruction prompts and results are given in Appendix D but the difference is not significant. The implementation details and the results of more baselines (Tu et al., 2023b;d) are given in Appendix B.

Table 3: Comparison of the methods fine-tuned on IER training set with and without *DiffTell*. The testing sets are the IER testing set and the PSBattle dataset.

| Testing Set | Method | *DiffTell* | BLEU@4 | METEOR | CIDEr | ROUGE-L |
|---|---|---|---|---|---|---|
| IER | CLIP4IDC | ✗ | 5.65 | 10.23 | 22.52 | 28.95 |
| | | ✓ | **8.64** | **13.54** | **28.14** | **36.84** |
| IER | OpenFlamingo-3B | ✗ | 4.45 | 14.87 | 15.80 | 29.79 |
| | | ✓ | **6.49** | **16.68** | **21.04** | **31.36** |
| IER | Fuyu-8B | ✗ | 4.85 | 11.84 | 23.67 | 28.10 |
| | | ✓ | **9.59** | **16.52** | **41.05** | **35.44** |
| IER | Llave-Interleave-8B | ✗ | 6.09 | 14.05 | 29.69 | 32.67 |
| | | ✓ | **11.06** | **17.35** | **44.79** | **37.21** |
| PSBattle | CLIP4IDC | ✗ | 0.00 | 3.08 | 1.59 | 13.83 |
| | | ✓ | **3.08** | **6.25** | **3.63** | **21.22** |
| PSBattle | OpenFlamingo-3B | ✗ | 2.35e-04 | 2.33 | **7.71** | **19.24** |
| | | ✓ | **2.12** | **6.60** | 4.02 | 16.10 |
| PSBattle | Fuyu-8B | ✗ | 1.38 | 4.79 | **4.19** | 12.23 |
| | | ✓ | **2.15** | **7.57** | 4.05 | **13.73** |
| PSBattle | Llave-Interleave-8B | ✗ | 2.60 | 8.88 | 7.86 | 18.01 |
| | | ✓ | **4.13** | **9.39** | **8.55** | **21.09** |

Table 4: Results of IER testing set from OpenFlamingo-3B model finetuned on different datasets.

| Metrics | IER | + InstructP2P | + OCR | + MagicBrush | + COCO | + GIER | + *DiffTell* |
|---|---|---|---|---|---|---|---|
| B@4 | 4.45 | 5.41 | 6.24 | 5.70 | 4.67 | 6.35 | **6.49** |
| M | 14.87 | 14.54 | 15.73 | 13.94 | 11.64 | 10.86 | **16.68** |
| C | 15.80 | 15.69 | 17.29 | 15.71 | 11.50 | 19.07 | **21.04** |
| R-L | 29.79 | 30.38 | 31.28 | 29.05 | 26.20 | 29.76 | **31.36** |

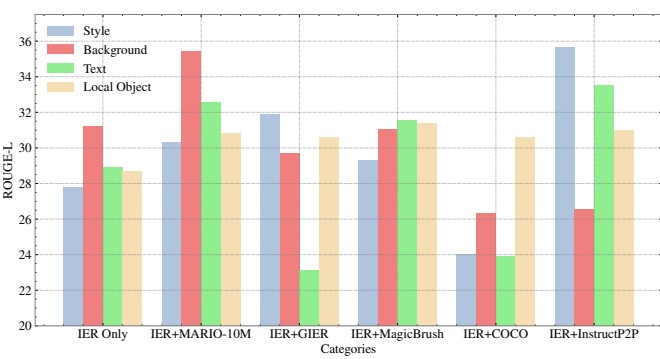 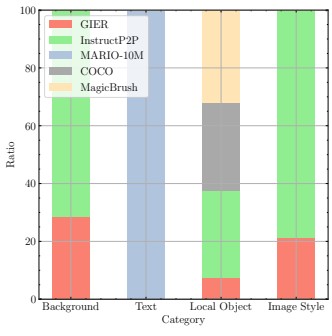

(a) Category-wise ROUGE-L comparison on IER testing set using OpenFlamingo-3B trained with different subsets in *DiffTell*.

(b) General statistics of the contribution to the difference categories from each subset in *DiffTell*.

Figure 3: Category-wise ROUGE-L score in *DiffTell* and general statistics of *DiffTell*

## 4.2 MAIN RESULT

**Quantitative Result** We report the experiment results on the IER testing set and PSBattle dataset with and without *DiffTell* in Table 3. Results demonstrate *DiffTell*'s ability to enhance performance across nearly all evaluation metrics and for all baseline methods, underscoring the contribution of the *DiffTell* dataset on IDC. Notice that OpenFlamingo-3B with LLM backbone is less capable than CLIP4IDC with a much smaller model size. We suspect that the Flamingo model does not have direct modeling of the interaction between the two images because each image feature is cross-attentioned by language token, then the language tokens will interact via causal attention. In contrast, in CLIP4IDC, the two image patch features extracted by CLIP are fused using a transformer, which is a direct information interaction among image tokens, serving as a strong condition to guide the transformer decoder to generate the language that describes the visual difference. There is no image encoder in the Fuyu model, and the image is patched linearly to the transformer. Thus, Fuyu can accept an image of the arbitrary size, improving its capability to detect tiny differences and small objects. This can be the reason why Fuyu improves greatly after fine-tuning. For Llava-Interleave-8B, the pre-trained interleaved dataset provides a good knowledge base for the model to understand the context with multiple image inputs. Thus, it outperforms the IDC-specific model without *DiffTell* and can perform best among all the baselines. In addition, the performance on the PSBattle dataset is generally lower than IER, which is as expected since PSBattle is used for zero-shot tests without the training set.

**Qualitative Study** We compare the prediction for the OpenFlamingo-3B and CLIP4IDC models trained with and without *DiffTell*. The visualization examples of IER and PSBattle testing set are shown in Figs. 4 and 5, respectively.

As depicted in Fig. 4, the model demonstrates enhanced proficiency in describing local object changes, text detection and recognition, background alterations, and image style changes. In the text manipulation example, the model exhibits OCR capabilities without relying on existing OCR techniques. Notably, in the local object change example, the model accurately identifies the addition of a tattoo on the girl's back, showcasing its capability to recognize modified objects and discern correct object relationships. Furthermore, in the third example depicting a background change, the model with *DiffTell* uses *around* rather than *from*, underscoring its spatial recognition capability.

In the zero-shot testing scenario of PSBattle, anticipating imperfect predictions is reasonable. However, it is crucial to observe the conceptual similarity between predictions and ground truth. Similar to the earlier findings, the model acquires the capability of object change perception and OCR even without an LLM backbone.

## 4.3 ABLATION STUDY

Since *DiffTell* is a dataset with several subsets contributing to different image difference categories, it is necessary to study the contribution of each subset to the IDC performance. We consider two parts: the contribution of each subset to the general performance and the contribution of each subset

<Local object change>  <Text manipulation>  <Background change>  <Image style change>

**Ours:** add an tribal tattoo to the girls *back*.
**W/o DiffTell:** remove the background
**GT:** color her skin darker, color her tattoo more black

**Ours:** add the text *Pool Party*.

**W/o DiffTell:** Change the background to blue.
**GT:** Add text "Pool Party

**Ours:** remove *all* the background *around* the cat.
**W/o DiffTell:** remove the background *from* the cat.
**GT:** Remove all background except for the cats face

**Ours:** change the background of the image to a gold color.
**W/o DiffTell:** change the background to a city.
**GT:** Change blue color to yellow

Figure 4: Visual comparison that illustrates the impact of utilizing the *DiffTell* dataset on Flamingo's performance across four distinct categories in the IER testing set. Our dataset demonstrates its effectiveness in enhancing performance, especially in local object description, text detection and recognition, spatial recognition, and image style description. The text in green shows an obviously precise expression over the text in red.

<Local object change>          <Text manipulation>

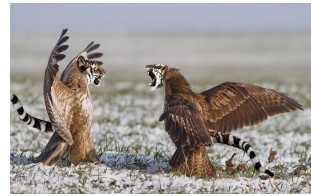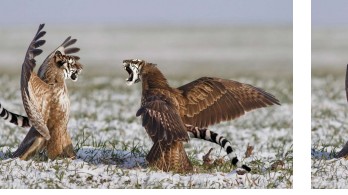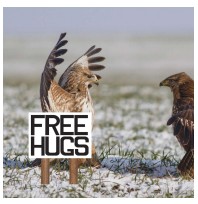

**Ours:** add the eagle has been replaced by a tiger.
**W/o DiffTell:** add a blue filter.
**GT:** The heads of the birds have been replaced with the heads of tigers.

**Ours:** add the text free hugs.
**W/o DiffTell:** add a blue filter  .
**GT:** The bird on the left now has a sign that says "Free Hugs" on it.

Figure 5: The visual comparison illustrates the impact of utilizing the DiffTell dataset on the CLIP4IDC model's performance across two categories in the PSBattle dataset.

to each category. We show the performance on the IER testing set from the OpenFlamingo-3B model finetuned with the IER training set and each subset in *DiffTell* in Table 4. **Almost every subset can improve the performance, and in sum, the *DiffTell* can boost the performance further.** The improvement is relatively marginal for the COCO dataset. One possible reason is the disparity in the data distribution. Only 23 categories of objects from the COCO dataset exist in the IER dataset, and COCO's caption template is not the same as that in the IER testing set.

We show another ablation study on the category-wise contribution. To better study the performance of each category, we compute the statistics of the IER testing set based on the category given in Section 3.2. The statistics are given in Table 6 in the Appendix. Fig. 3a provides an overview of the contributions based on the IER testing set and OpenFlamingo-3B of each subset in *DiffTell* to each category, regarding ROUGE-L. For detailed results across all categories and evaluation metrics, please refer to the Appendix F. Compared to the model trained exclusively on IER, the model trained on our subset derived from MARIO-10M shows a notable performance improvement, benefiting from the versatility of words in various real-life scenarios. Our subset derived from GIER contributes positively to overall performance, except for text manipulation, where no such data exists in the GIER dataset. The absence of background change data in the MagicBrush dataset leads to a performance decrease in the background change category. COCO, designed for local object changes, enhances performance in this category. In the InstructPix2Pix dataset, the lack of background modification data results in a performance decrease, specifically in background change. **In summary, the subset belonging to the specific categories can generally contribute to the corresponding categories in the IER testing set.**

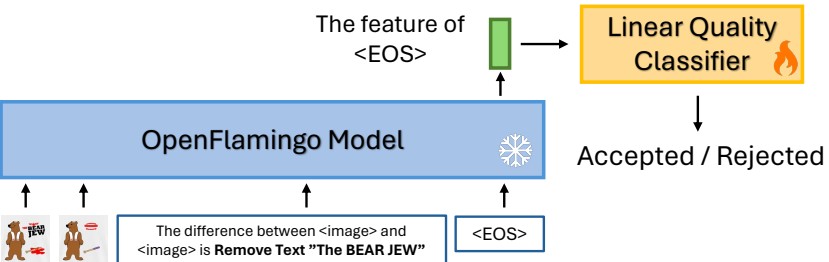

Figure 6: The framework of the automatic data filtering pipeline. The image pair and difference caption will be passed to the OpenFlamingo model, and the output feature of <EOS> token will be used for the classification of acceptance or rejection

Table 5: The results of performance on IER testing set using the data with automatic classifier or not.

| Training Set | BLEU@4 | METEOR | CIDEr | ROUGE-L |
|---|---|---|---|---|
| IER | 4.45 | 14.87 | 15.80 | 29.79 |
| IER + 10K random Data | 4.41 | 14.88 | 15.59 | 29.63 |
| IER + 10K data filtered by classifier | 6.01 | 15.41 | **17.66** | 31.08 |
| IER + 10K filtered by the human | **6.10** | **15.54** | 17.39 | **31.11** |

## 4.4 AUTOMATIC DATA FILTERING

The cost of manual data filtering can become a bottleneck when scaling up this dataset. To address this, we propose an alternative automatic data filtering pipeline, as shown in Fig. 6. Using a dataset previously reviewed by humans, we compile both accepted and rejected samples as the training set for a binary classifier. The classifier's input consists of features extracted by the OpenFlamingo-3B model, which has been fine-tuned on the IDC task. This classifier can assist annotators in more efficiently filtering the data.

To validate the effectiveness of our pipeline, we train a quality classifier on an annotator-validated subset of MARIO-10M, comprising 10K accepted and 10K rejected samples. We use an SVM as the classifier, splitting 16K samples for training and 4K for testing, achieving an accuracy of 85.22%. The classifier is then applied to unseen data from MARIO-10M, filtering 10K accepted samples. This unseen data is newly in-painted using FireFly Generative Fill, as explained in Section 3.3, and generation stops once 10K accepted samples are collected through the classifier. We compare the performance on IER dataset of the IDC model (OpenFlamingo-3B) trained on three subsets from MARIO-10M: 10K auto-filtered samples, 10K randomly selected samples, and 10K manually filtered samples. The randomly selected data is taken directly from the in-painted model without quality control, while the manually filtered data is a subset of MARIO-10M used in *DiffTell*. The results in Table 5 demonstrate that the auto-filtered training data can achieve much better performance than unfiltered data (random data), and be comparable to human filtered training data. Such a result shows the necessity of the filtering step in our designed pipeline and highlights the classifier's effectiveness and the potential for scaling data collection using this auto-filtering pipeline.

## 5 CONCLUSION AND LIMITATION

In this study, we introduce *DiffTell*, an extensive and high-quality dataset for image difference captioning (IDC). This dataset addresses the gaps in diversity and scale that were previously present in the IDC task. Through comprehensive experiments conducted on diverse testing sets and employing various baseline methods, we demonstrate the efficacy of our dataset in enhancing performance. Additionally, we analyze to understand the improvement contributed by each component of *DiffTell* to different image difference categories. We aspire that *DiffTell* will play a significant role in advancing the development of more sophisticated multi-modality models for IDC and language-guided image editing in the future. As for future work, at this time, we only use human-filtered data for supervised fine-tuning. We hope to utilize the human-filtered data (acceptance and rejection) for preference optimization (Rafailov et al., 2024; Meng et al., 2024) to boost the performance.

## ETHIC STATEMENT

This work does not involve potential malicious or unintended uses, fairness considerations, privacy considerations, security considerations. We claim to adhere the Code the Ethics.

## REPRODUCIBILITY STATEMENT

We provide details to reproduce our results in Appendix B. The data collection details are given in Section 3.3 and Appendix G. We will release the code and the dataset upon acceptance.

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

**RoadMap** The supplementary matrial is composed as follows. Section A presents a detailed description of the existing datasets in IDC. Section B gives the implementation details. Section C presents more baselines which are not included in the main paper. Section D presents the details using diverse instruction prompts. Section E presents the zero-shot or few-shot performance on LLMs without being finetuned on IER testing set. Section F presents more results from ablation study. Section G discusses more about the dataset collection. Section H gives a brief introduction of PSBattle dataset. Section I illustrate some failure cases. Section J discusses the limitation. We will release the code and the dataset once the paper is accepted.

## A    EXISTING DATASETS

The most commonly used datasets in the IDC task are CLEVR change (Park et al., 2019b), Spot-the-Diff (Jhamtani & Berg-Kirkpatrick, 2018b) and Image Editing Request (IER) (Tan et al., 2019). CLEVR change constitutes a sizable synthetic dataset characterized by moderate viewpoint variations. Spot-the-difference is composed of pairs of frames extracted from video surveillance footage and the corresponding textual descriptions of visual changes. IER is crawled from the practical image editing requests from the Reddit channel, consisting of 3,939 pairs of real images, accompanied by 5,695 editing instructions. Each image pair in the training set is associated with one instruction. In contrast, each image pair is linked to three instructions for a more objective evaluation in the validation and testing sets. Because IER is collected from a real-world scenario, it covers more image difference categories such as background change, text manipulation, and local object change. The definition of the image difference categories can be found in Section 3.2. Due to the single domain in CLEVR and Spot the Difference datasets, we mainly use IER in this work as the testing set, which aligns our scope to have a comprehensive, diverse, and practical dataset.

Table 6: Statistics of each image difference category in the IER testing set.

| Category | Background | Text | Local object | Image style |
|---|---|---|---|---|
| Number of Images | 117 | 53 | 277 | 223 |

## B    IMPLEMENTATION DETAILS

### B.1    TRAINING DETAILS

For CLIP4IDC, We adopt the official implementation of CLIP4IDC. However, as it lacks the training script and the pretrained weights for IER, we reproduce the CLIP4IDC[4] model trained on IER exactly following its provided training hyper-parameter settings of the CLEVR dataset. For VARD-LSTM[5] and NCT[6], there is still no official implementation for IER and we reproduce them using the settings in CLEVR dataset. The pre-trained Biaffine Parse in NCT we use is from Diaparser[7]. For OpenFlamingo-3B, the vision encoder and language encoder are `ViT-L-14` and `anas-awadalla/mpt-1b-redpajama-200b`. The cross attention interval is 1. For OpenFlamingo-9B, the vision encoder keeps the same and the language encoder becomes `anas-awadalla/mpt-7b`. The cross attention interval is 4. For LLaVA-Interleave-8B, the language model we use is `meta-llama/Meta-Llama-3-8B-Instruct`. For Fuyu-8B, we use `adept/fuyu-8b`. The training platform we use is 8 NVIDIA A100s with the 80GB GPU memory. The training epochs is 10 for the MLLMs and the base learning rate is $1e-5$ with cosine scheduler. The weight decay is 0.01 and the global batch size 128. The training will last about 20 hours.

---

[4]https://github.com/sushizixin/CLIP4IDC
[5]https://github.com/tuyunbin/VARD
[6]https://github.com/tuyunbin/NCT
[7]https://github.com/Unipisa/diaparser

## C  THE PERFORMANCE OF MORE BASELINES

Besides the methods in the main text, we test more baselines including NCT (Tu et al., 2023e) and VARD-LSTM (Tu et al., 2023b) given in Table 7.

Table 7: The comparison of the methods fine-tuned on image editing request (IER) training set with and without *DiffTell* using more baselines.

| Testing Set | Method | DiffTell | BLEU@4 | METEOR | CIDEr | ROUGE-L |
|---|---|---|---|---|---|---|
| IER | NCT | ✗ | 1.64 | 7.97 | 7.47 | 19.40 |
| | | ✓ | **1.94** | **9.63** | **7.58** | **23.79** |
| IER | VARD-LSTM | ✗ | 1.60 | 8.06 | 5.49 | 18.87 |
| | | ✓ | **1.71** | **8.54** | **6.02** | **20.08** |
| PSBattle | NCT | ✗ | 2.78e-08 | 0.73 | 1.12 | 4.53 |
| | | ✓ | **1.65e-06** | **1.22** | **3.11** | **9.78** |
| PSBattle | VARD-LSTM | ✗ | 1.49e-08 | 0.43 | 1.56 | 7.01 |
| | | ✓ | **7.46e-07** | **0.88** | **2.07** | **7.79** |

## D  THE EXPERIMENTS WITH DIVERSE PROMPTS

In instruction tuning, incorporating diverse prompts enhances model robustness, making them more adaptable and better at generating accurate responses across varying contexts (Bukharin & Zhao, 2023). Initially, we use a uniform prompt "What is the difference between two images?" across all datasets and ask the model to provide an answer. To ablate this, we expand the prompt into nine different variations and compare the performance against the single-prompt approach, as shown in Table 8. The nine prompts we use are as follows. The model we use is OpenFlamingo-3B. As a complex vision-language task, it is more important for the model to understand two images, identify the difference and express the answer. Thus, to improve the vision encoder could be more useful.

• Please tell me the editing instruct of how to edit <|image|> to look like <|image|>.

• Identify the transformations applied to <|image|> to achieve the appearance of <|image|>.

• Outline the steps required to edit <|image|> so that it matches the look of <|image|>.

• Explain the edits necessary to convert <|image|> into <|image|>.

• What alterations were made to <|image|> to create <|image|>?

• Detail the changes from <|image|> to <|image|>.

• <|image|> is image1, <|image|> is image2, tell me what the change is between these two images.

• <|image|> is image1, <|image|> is image2, tell me what the change is from image1 to image2.

Table 8: The results of performance on IER testing set using the diverse prompts. The model we use is OpenFlamingo-3B.

| Testing Set | DiffTell | D. Prompt | BLEU@4 | METEOR | CIDEr | ROUGE-L |
|---|---|---|---|---|---|---|
| IER | ✗ | ✗ | 4.45 | 14.87 | 15.80 | 29.79 |
| | ✓ | ✗ | **6.49** | **16.68** | 21.04 | **31.36** |
| | ✓ | ✓ | 6.32 | 16.59 | **23.88** | 30.34 |
| PSBattle | ✗ | ✗ | 2.35e-04 | 2.33 | **7.71** | **19.24** |
| | ✓ | ✗ | **2.12** | **6.60** | 4.02 | 16.10 |
| | ✓ | ✓ | 1.77 | 6.45 | 4.48 | 16.46 |

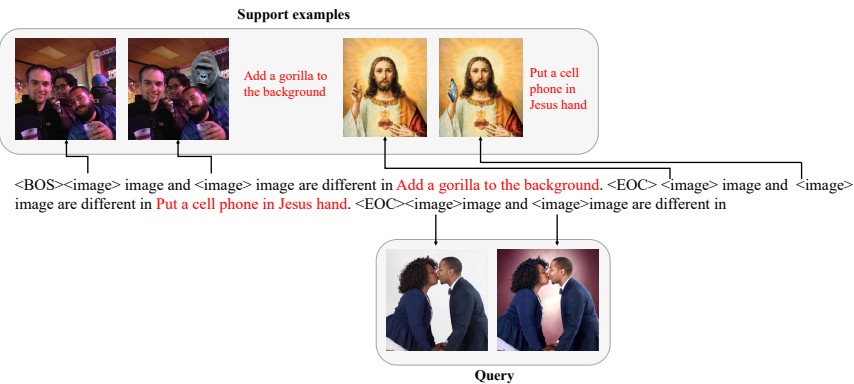

Figure 7: The example of how we construct the few-shot prompt.

Table 9: The results of zero-shot or few-shot prompt results on IER testing set. The few-shot prompt is the composition of 3 training examples from the training set.

| Method | Few-shot | BLEU@4 | METEOR | CIDEr | ROUGE-L |
|---|---|---|---|---|---|
| OpenFlamingo-3B | ✗ | 1.18 | 8.07 | 8.72 | 16.63 |
| | ✓ | 0.84 | 7.64 | 4.09 | 17.54 |
| OpenFlamingo-9B | ✗ | 1.15 | 8.26 | 6.04 | 19.00 |
| | ✓ | 1.99 | 9.18 | 5.01 | 20.93 |

## E  ZERO-SHOT/FEW-SHOT PROMPT RESULTS

Investigating the potential of zero-shot learning is essential for the method utilizing LLM. For few-shot prompt testing, we randomly choose three examples from the IER training set. Performance results on the PSBattle dataset are not reported due to the absence of training data in that specific dataset. The detailed results can be found in Table 9. The few-shot prompt example is shown in Fig. 7. The results show that image difference caption (IDC) is a hard task for the current LLMs although they are trained on huge amount of data. Even with few-shot prompt, the results are still not satisfying.

## F  MORE ABLATION STUDY RESULTS

Due to the page limit in the main paper, we only present the contribution of each subsets of *DiffTell* to the performance of each category regarding ROUGE-L in Fig. 3a. We present the other three metrics here as shown in Figs 8, 9, 10, respectively.

Based on the four evaluation metrics, we can find that each dataset can contribute to at least one category of the performance on IER, showing that the positive effect by enlarging the dataset, which is the aim of this work.

## G  DATASET COLLECTION DETAILS

**Image In-painting**   We use FireFly Generative Fill to in-paint the image. The inputs we can provide are the original image and the prompt for the generative model. There is no need for us to select the parameters. The illustration is given in Fig. 11. We generate $I_2$ for COCO and MARIO-10M subsets in *DiffTell*.

**Data Filtering**   The illustration of how the annotators filter the data is given in Fig. 12 , Fig. 13 and Fig. 14 which are for InstructPix2Pix, COCO and MARIO-10M subsets, repsectively. For InstructPix2Pix, the annotators filter whether the $T_{I_1, I_2}$ matches $(I_1, I_2)$ or whether the change

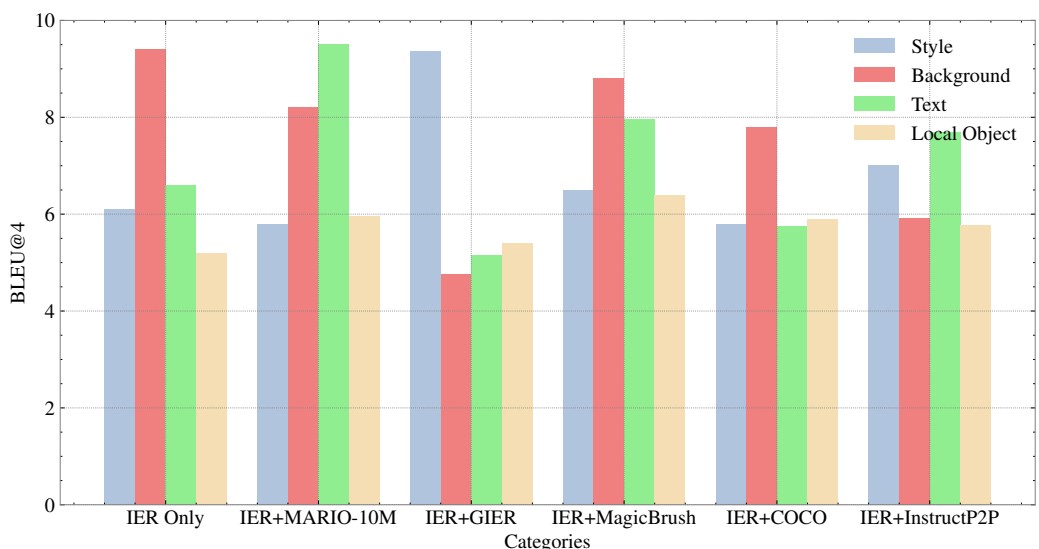

Figure 8: The category-wise BLEU@4 comparison on IER testing set using OpenFlamingo-3B trained with different subsets in *DiffTell*.

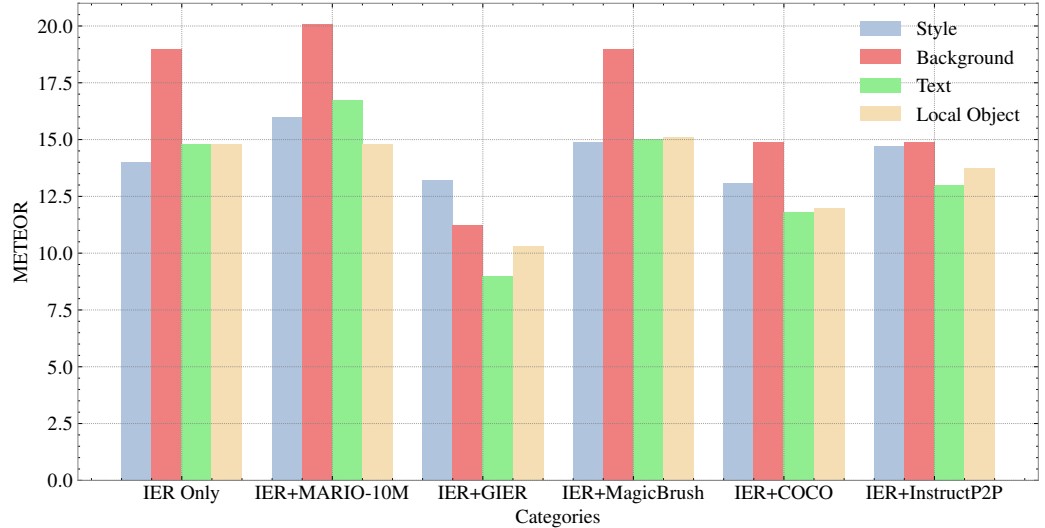

Figure 9: The category-wise METEOR comparison on IER testing set using OpenFlamingo-3B trained with different subsets in *DiffTell*.

reflects on $I_1$ and $I_2$ because $(I_1, I_2, T_{I_1, I_2})$ has already been provided. For COCO and MARIO-10M only providing $I_1$, the annotators filter whether the object or the text is successfully in-painted from $I_1$.

## H PSBATTLE DATASET

The PSBattle dataset is another practical dataset used in (Black et al., 2024) that consists of images edited in Adobe Photoshop™ and is curated from the "Photoshopbattles" subreddit. We include this dataset only for the evaluation of out-of-domain data to test the generalizability of the models. This dataset comprises over 10,000 images, each paired with several modified variants generated according to editing instructions provided by users. In total, there are 102,208 variants created by 31,000 different artists. For our study, we randomly selected 100 image pairs, each accompanied by

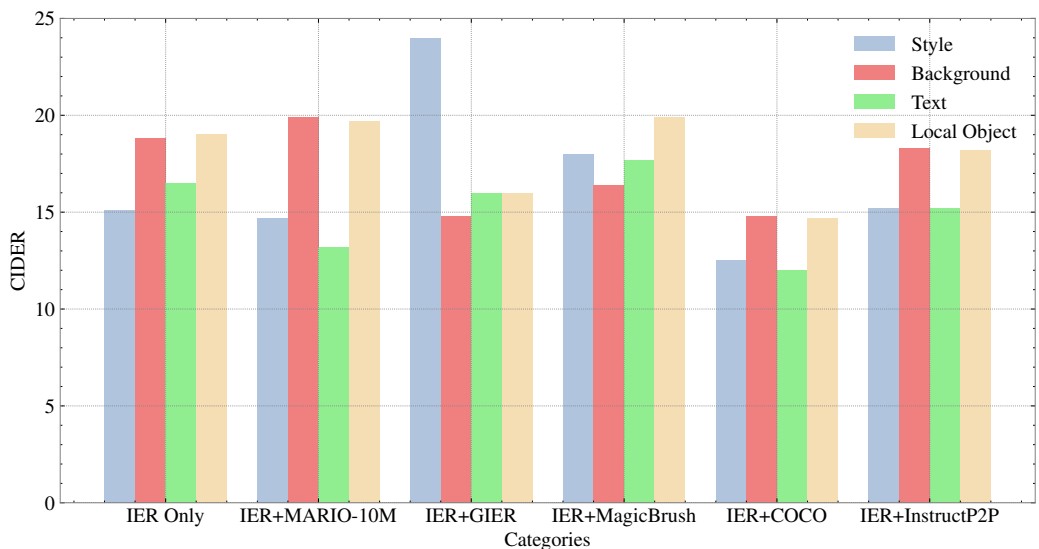

Figure 10: The category-wise CIDER comparison on IER testing set using OpenFlamingo-3B trained with different subsets in *DiffTell*.

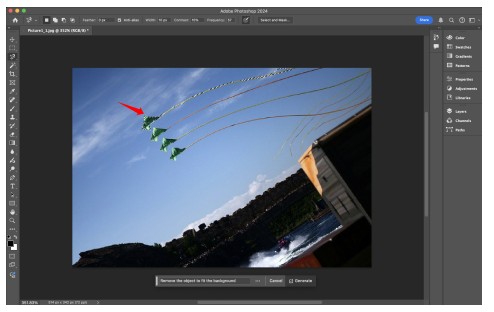
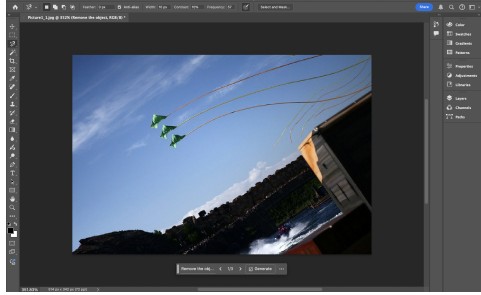

(a) The image before in-painting        (b) The image after in-painting

Figure 11: We in-paint the image using Firefly Generative Fill in PhotoShop. For each image, we provide the original image ($I_1$) and the corresponding mask. The mask is used to identify the selected area shown with the red arrow. We use prompt to ask Firefly to in-paint the image and fit the background. Normally, the Firefly will return 3 to 4 in-painted images.

three captions obtained through crowd-sourced annotation on MTurk. The illustration of PSBattle dataset is shown in Fig. 15.

# I   FAILURE CASES

Although the model gains performance improvement in IDC, there are still some cases where the model fails to predict correctly. We illustrate the failure cases in Fig. 16. The model may sometimes limit its predictions to local changes rather than providing a comprehensive description. In the first example shown in Fig. 16, the model exclusively identifies the difference in the head from the body, neglecting the other face and the relationship between the two faces. Although the model recognizes the change in the second example, it produces an inaccurate description. These shortcomings may result from the limited diversity in the dataset. A predominant portion of the images in *DiffTell* originates from real-life scenarios. The model struggles to capture surreal or fantastical compositions, such as a body with two heads, as the training data may not adequately represent those instances. Following our methodology in creating *DiffTell*, incorporating more data sources covering a wider range of fine-grained domains may help the model to establish connections between objects and

**Instructions:** Given an input image, the output image and the editting instruction. The meanings of the terms are as follows:

- Input Image: The original image we want to edit.
- Output Image: The image generated by AI model based on the input image.
- Editting Instruction: The instruction used to guide the AI model to generate the output image from the input image.

You are supposed to evaluate whether the ouput image is matched with the input image and the editing instruction. After carefully check the images and the instruction, you should select the quality score for the output image. Please check the Yes for the successful editting while No for the unacceptable editting.

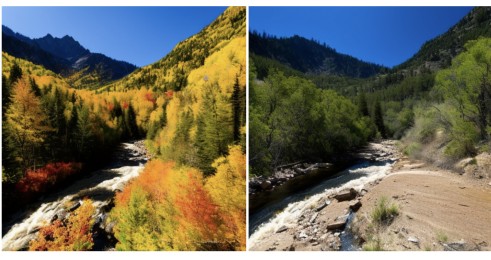

input image                    output image

**Editting Instruction:** *make the creek dry*

Figure 12: The labeling illustration of InstructPix2Pix subsets. The two images are $I_1$ and $I_2$. $T_{I_1, I_2}$ is given in **Editing Instruction**. The annotator is asked to identify whether the $T_{I_1, I_2}$ matches $(I_1, I_2)$ or whether the change reflect on $I_1$ and $I_2$ and give the answer "Yes" or "No". We keep those which are identified as "Yes".

**Instructions:** Given an input image, the input mask and a object-free image. The meanings of these 3 images are as follows:

- Input Image: The original image we want to remove the object.
- Input Mask: The region of the object generated by AI model. Ideally the mask should cover the object we want to remove.
- <object>-free Image: The image processed by AI model. **Ideally, there should not exist the <object> covered the mask and no extra element should be added. The <object> here is a placeholder which be will replaced by a specific object word.**

You are supposed to evaluate the object-free image, whether the object is fully removed without changing the original image content. After carefully compare the object-free image and the input image, you should select the quality score for how well the object is removed. We set 2 levels regarding the quality of the object-free image which are:

- Acceptable
- Unacceptable

The detailed criterion for the 2 categories and the corresponding example are given in the instrcution document.

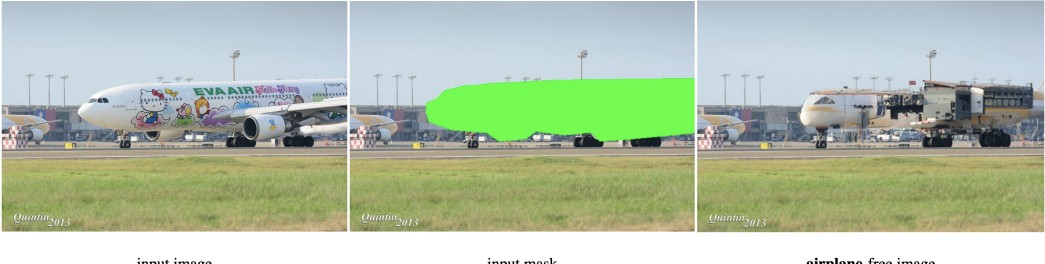

input image                    input mask                    **airplane**-free image

Figure 13: The labeling illustration of COCO subsets. From the left to the right, the first, second and third images are the original image ($I_1$), the input mask and the in-painted image. We provide the input mask and object name to remind the annotator which area should focus on. The annotator selects "Acceptable" and "Unacceptable". We keep those which are identified as "Acceptable".

**Instructions:** Given an input image, the input mask and a text-free image. The meanings of these 3 images are as follows:

- Input Image: The original image we want to remove the text.
- Input Mask: The region of the text generated by AI model. Ideally the mask should cover all the text.
- Text-free Image: The image processed by AI model. **Ideally, there should not exist text and no extra element should be added.**

You are supposed to evaluate the text-free image, whether the text is fully removed without changing the original image content. After carefully compare the mask-free image and the input image, you should select the quality score for how well the text is removed. We set 2 levels regarding the quality of the object-free image which are:

- Acceptable
- Unacceptable

The detailed criterion for the 2 categories and the corresponding example are given in the instrcution document.

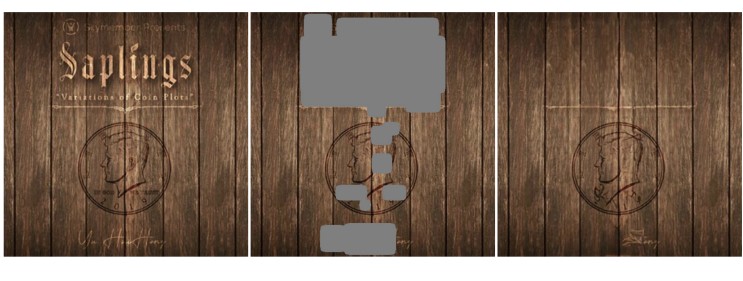

input image         input mask         text-free image

Figure 14: The labeling illustration of MARIO-10M subsets. From the left to the right, the first, second and third images are the original image ($I_1$), the input mask and the in-painted image. We provide the input mask and object name to remind the annotator which area should focus on. The annotator selects "Acceptable" and "Unacceptable". We keep those which are identified as "Acceptable".

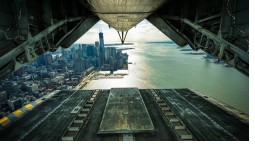 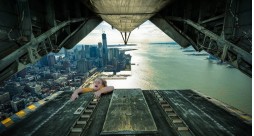 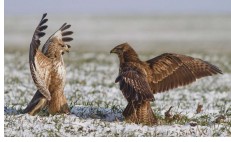 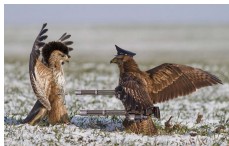

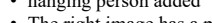

- hanging person added
- The right image has a person hanging off the end of the track with a horrified expression on his face.
- On the right, a man is clinging to the bomb bay door, about to fall. He is not there at all on the left.

- In the right picture the gun is visible
- Added Head hair in left eagle and cap and gun in the left one.
- Hawks are fighting each others in second one Hawk kept machine gun.

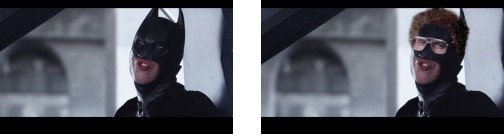 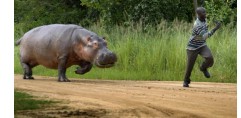 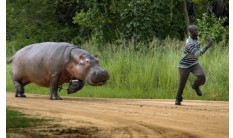

- A new face has been given to batman. I think it is the face of Will Ferral.
- The mask only covers part of the face and the man wears glasses now.
- Batman has been given a bushy head of hair and a large pair of glasses.

- The hippo is wearing a cross and holding a bible.
- The hippo is now carrying a bible and a crucifix necklace.
- The hippo is holding a bible and a crucifix in one of its hooves.

Figure 15: Four examples in PSBattle dataset.

accurately identify specific object categories, thus providing detailed captions for cases like the object in the tattoo.

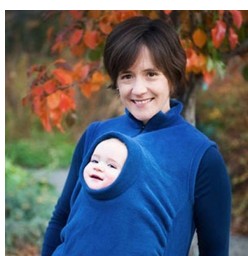 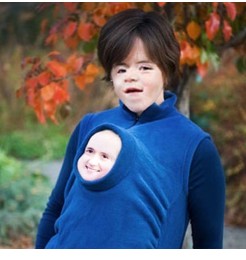 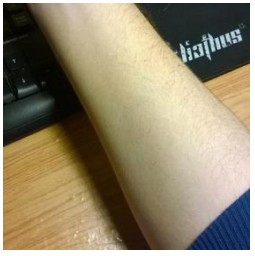 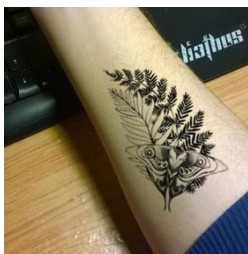

**Ours:** the woman has been turned into a boy.

**GT:** The baby face and the woman's face have been altered in the new one.

**Ours:** add bird.

**GT:** Add butterfly fern tattoo image.

Figure 16: Illustration of the failure cases from the model trained with DiffTell. The first example is from CLIP4IDC on PSBattle. The second is from Flamingo on the IER testing set.

## J    MORE VISUAL RESULTS

### J.1    FAILURE CASES IN DATA FILTERING

As mentioned in Section 3.3, we present the importance of the data filtering by showing more cases in InstructP2P, COCO and MARIO-10M datasets in Figure 17, 18 and 19, respectively.

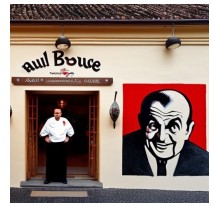 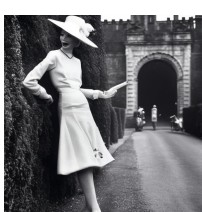 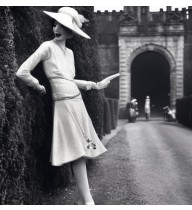

*make the mural of the Michelin Guide*                    *make it 1920s*

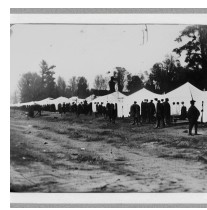 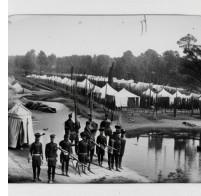 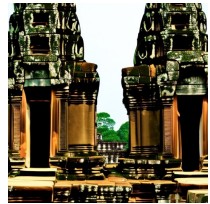 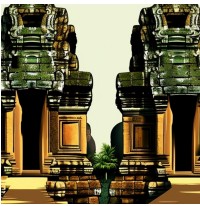

*Add a river.*                    *as a cartoon*

Figure 17: Within the InstructP2P dataset, we have identified four sets of images, each composed of the original image, the altered image, and the corresponding instruction. All four of these image sets represent instances of failure. In the first pair of images, not only is the mural altered as per the instruction, but there are also changes to the face of the person in white and the text on the wall. The second pair exhibits subtle changes that are unrelated to the provided instruction. For the third pair, the images undergo significant alterations, including the addition of a river, surpassing the intended modifications. In the fourth pair, the changes between the two images fail to accurately reflect the given instruction. The InstructP2P dataset is characterized by a high noise ratio, leading to a low acceptance rate of 35.13% during manual filtering.

COCO Image  COCO Mask  In-painted Image

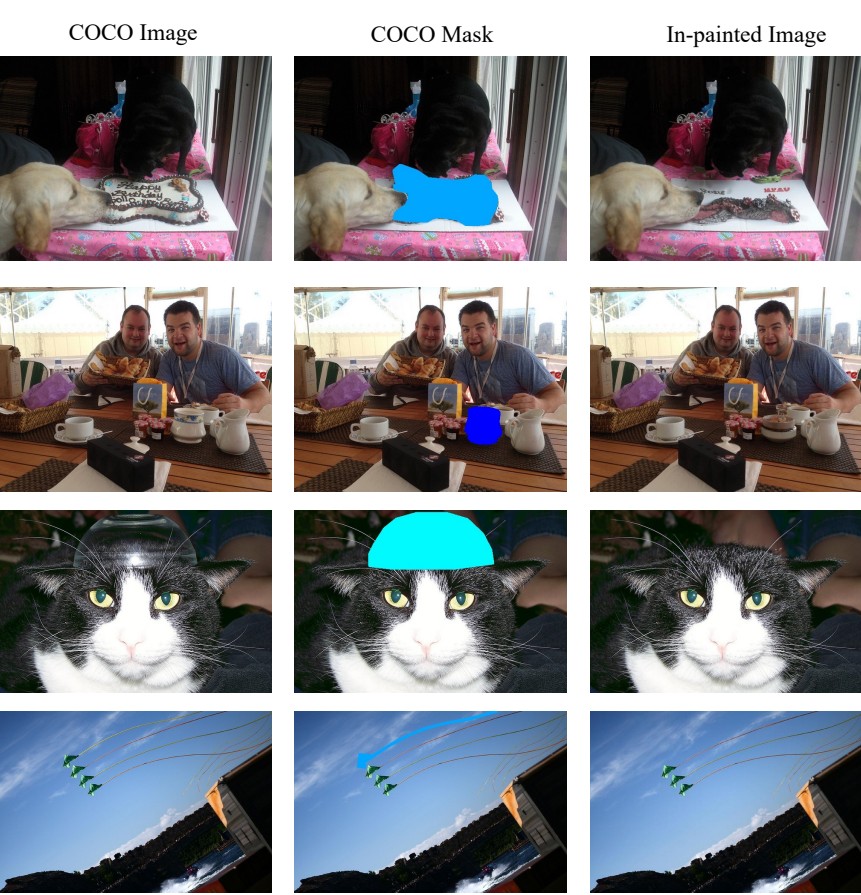

Figure 18: We choose four sets of images in COCO dataset, each comprising the original image, the dilated mask, and the in-painted image. The initial two sets depict instances of failure, whereas the latter two sets showcase successful outcomes. The initial failure occurs when the mask fails to adequately cover the object, and the second failure is attributed to the inadvertent addition of another object despite the mask effectively covering the intended object. The labeling team is instructed to exclude images falling into *DiffTell*.

MAIRO-10M Image     MAIRO-10M Mask     In-painted Image

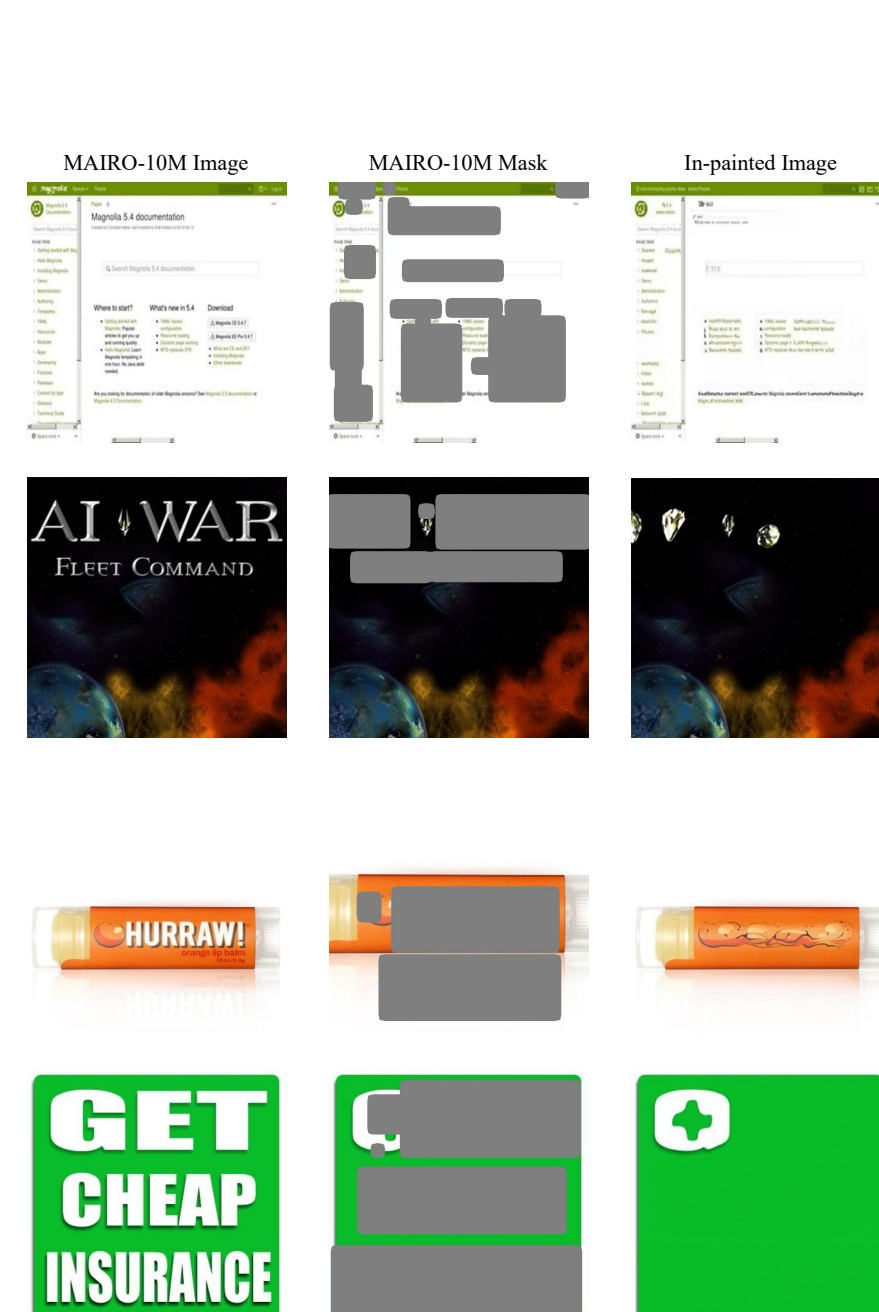

Figure 19: We select four image sets from the MAIRO-10M dataset, each including the original image, the dilated mask, and the in-painted image. All four of these cases have been deemed failures and subsequently excluded by the labeling team. The mask in MARIO-10M dataset is not very precise. All of these 4 image sets show this issue. In the first image set, the text is not very clear, either. Besides the inadequate mask and addition objects which exist in the COCO dataset, another issue of MARIO-10M dataset is the existence of low-quality images.

Figure 20: More examples from IER testing dataset regarding the four categories from OpenFlamingo-3B.

## J.2 More Successful Cases

To better illustrate the improvement from *DiffTell*, we select another two prediction results in IER testing set from the four categories respectively, shown in Fig. 20. The model we use is OpenFlamingo-3B.

