# OpenReview forum: "DiffTell: A Comprehensive Dataset for Image Difference Captioning"
_ICLR.cc/2025/Conference — Submitted to ICLR 2025_

### Official Review · Reviewer_nVjR · 2024-10-26

**Soundness:** 2
**Presentation:** 2
**Contribution:** 3
**Rating:** 5
**Confidence:** 4

**Summary:**

In contrast to prior image difference captioning (IDC) datasets that are limited in quantity or styles, this paper introduces a large-scale IDC dataset containing 4 types of image differences (background change, local object change, style change, and text manipulation) and both real and synthetic images. To collect such a comprehensive and extensive dataset, a two-step collection pipeline is implemented, where humans are involved to ensure data quality.
Empirical results show that models pretrained on this new dataset, DiffTell, achieve significant improvement across all evaluation metrics compared to models without such pretraining.

**Strengths:**

1. The proposed IDC captioning dataset, DiffTell, is more comprehensive and extensive than previous IDC captioning datasets.

**Weaknesses:**

1. The main concern of the proposed dataset is its image difference captions.
(a). The average length of the captions in DiffTell is 9.72 words only. Whether captions of this length can describe the image differences in detail remains a question.
(b). The diversity of the captions is another concern. A simple and fixed language template is used for generating descriptions of image pairs from the COCO and MARIO-10M datasets, which contribute over half of DiffTell’s data, potentially restricting the diversity of descriptions.

2. There is insufficient clarity in the experiments, and some results appear ambiguous.
(a). In Table 3, it’s unclear whether models are trained on DiffTell first and then finetuned on the IER training set or if they are trained on both DiffTell and the IER training set simultaneously.
(b). How does the model perform on the IER testing set in a zero-shot setting (i.e., finetune the model on DiffTell only)?
(c). In Figure 3 (a), the model trained with IER+MARIO-10M shows significant improvements across all categories compared to those trained on IER alone, but MARIO-10M provides Text category data only. Where does improvement in other categories come from? Similarly, InstructP2P contributes data for the Background category and none for Text, but the model trained with IER+InstructP2P improves significantly in Text but performs worse in Background. Is there any further explanation for this?
(d). In Table 9, it's unclear why OpenFlamingo-3B performs worse in the few-shot setting than in the zero-shot setting.

**Questions:**

1. Why is the CLEVR dataset not considered in collecting DiffTell? Although CLEVR contains images in a single domain (toy bricks), it could enhance models’ 3D spatial understanding.
2. What model is used to generate masks for the COCO dataset?

---

### Official Review · Reviewer_PJb5 · 2024-11-03

**Soundness:** 3
**Presentation:** 3
**Contribution:** 3
**Rating:** 6
**Confidence:** 3

**Summary:**

This paper proposes a new dataset to boost existing methods on Image Difference Captioning tasks. The images in the dataset are collected from publicly available sources, while the annotations are human-filtered.

**Strengths:**

The DiffTell dataset is a large dataset for the Image Difference Captioning (IDC) task, which considers various types of differences between images. After incorporating DiffTell dataset in training process, existing methods can achieve better performance on IDC tasks.

**Weaknesses:**

As the CLEVR-Change and DiffTell datasets are of a similar scale, both of them containing 70k samples, the paper should include a comparative analysis of models trained on these two datasets.

According to Figure 3(b), the subset that considers differences in text is entirely from MARIO-10M. However, as shown in Figure 3(a), the model trained on IER+InstructP2P achieves higher performance on captioning difference on text than the model trained on IER+MARIO-10M. The paper should provide an analysis of this discrepancy.

Additionally, the “i” in the first “image” in the first sentence of the abstract should be capitalized.

**Questions:**

Please see the questions in Weaknesses.

---

### Official Review · Reviewer_Adcx · 2024-11-04

**Soundness:** 2
**Presentation:** 3
**Contribution:** 2
**Rating:** 5
**Confidence:** 4

**Summary:**

This paper introduces the DiffTell dataset for the Image Difference Captioning (IDC) task, which involves describing the distinctions between two images. The existing datasets are noted to lack comprehensive coverage of various image-difference categories, prompting the creation of DiffTell. This dataset includes a wide range of differences, such as global image alterations, object-level changes, and text manipulations, and combines newly collected data with filtered data from prior studies. To efficiently scale data collection and maintain quality, the authors investigate automatic filtering methods. The study shows that training on the DiffTell dataset improves performance for both traditional methods and previous multimodal large language models in the IDC task.

**Strengths:**

1. The author's argumentation is well-structured, with logical flow and thorough  literature review.
2. The dataset constructed in this paper can advance the field of image difference analysis， in which  there are more comprehensive coverage of various image-difference categories.
3. Some experiments demonstrated the effectiveness of the constructed dataset.

**Weaknesses:**

1. While Tables 1 and 2 present comparisons between the proposed DiffTell dataset and previous related works, they do not include a comparison of the length of image difference captions, which is an important aspect to consider.
2. In Table 3, the authors include previous classic MLLM models as baselines, but it would be beneficial to supplement these baselines with more recent MLLM works to fully demonstrate the advantages of the DiffTell dataset for the image difference captioning task.
3. Figures 4 and 5 illustrate the model capability gains of the DiffTell dataset for achieving image difference captioning, but showcasing some failure cases would more clearly highlight the current limitations of the DiffTell dataset.
4. Although the paper shows performance improvements from using the data with an automatic classifier in Table 5, it does not quantify the accuracy of the automatic classifier in filtering the data.
5. This article focuses on the construction of a dedicated task dataset, while the task itself is not new, and no new methods are presented in the paper. From the perspective of innovation, the contribution is limited.

**Questions:**

1. The authors should provide a comparison of the lengths of image difference captions in the DiffTell dataset versus those in previous datasets.
2. It would strengthen my evaluation to include additional recent MLLM models as baselines in Table 3. Supplementing classic models with more contemporary ones would better showcase the advantages of the DiffTell dataset.
3. Adding examples of failure cases in Figures 4 and 5 would illustrate the limitations of the DiffTell dataset. This would provide valuable insights into areas for improvement and highlight contexts where the DiffTell dataset may struggle.
4. The performance improvements shown in Table 5 are compelling. However, quantifying the classification accuracy of the automatic classifier used for filtering the data is necessary. Understanding the classifier's effectiveness would enhance the credibility of the reported performance gains.

---

### Official Review · Reviewer_p9ho · 2024-11-04

**Soundness:** 2
**Presentation:** 3
**Contribution:** 1
**Rating:** 3
**Confidence:** 5

**Summary:**

This paper presents the DiffTell dataset, developed for the image difference captioning (IDC) task. DiffTell focused on image pairs that exhibit various manipulations, including both synthesized and Photoshopped images. The dataset incorporates four types of image differences: background change, local object change, text manipulation, and style change.

**Strengths:**

1. The dataset encompasses a diverse range of image modification types from various sources, which increases the dataset’s variability in modification categories.
2. DiffTell may have practical applications in detecting manipulated images and generating descriptive captions for them, may contributing to fields like image forgery detection and multimedia forensics.

**Weaknesses:**

1. The DiffTell dataset is limited to synthetic or edited images, whereas typical IDC tasks more often involve pairs of real images, as seen in datasets like Spot-the-Diff and Birds-to-Words. By narrowly defining the IDC task as one limited to manipulated images, the paper restricts its findings and contributions to only synthetic or Photoshopped cases, which may limit the general applicability of the conclusions in real-world scenarios.

2. The differences in image pairs within DiffTell, which rely entirely on manipulations, are more easily distinguishable, such as through pixel by pixel subtraction. This raises concerns that models trained on DiffTell could “cheat” by learning these manipulated differences rather than truly comparing two images. Thus, these models may not perform well in identifying nuanced differences in real image pairs.

3. The paper’s experimental validation is limited to IER and PSBattle datasets, both of which are also manipulation-focused. However, it excludes testing on more general IDC datasets like Spot-the-Diff or Birds-to-Words, which would offer insight into the model's efficacy with real-world image pairs.

4. This paper does not address prior work sufficiently, particularly the recent OneDiff study [1], which also explored IDC with a variety of data sources and employed multimodal large language models. A comparison of DiffTell with OneDiff in terms of contributions and distinctions is essential to demonstrate DiffTell's novel aspects and improvement over existing datasets.

[1] Hu, E., Guo, L., Yue, T., Zhao, Z., Xue, S., & Liu, J. (2024). OneDiff: A Generalist Model for Image Difference. arXiv preprint arXiv:2407.05645.

**Questions:**

1. Considering the limitations identified above, specifically the restricted focus on manipulated images, how does the proposed dataset and approach aim to generalize to IDC tasks involving real, unaltered image pairs? Would DiffTell-trained models require additional fine-tuning on real image pairs for effective real-world application?

2. Could the authors clarify how DiffTell fundamentally differs from OneDiff or other previous works, particularly in dataset construction, data quality, and IDC model performance?

---

### Meta-Review · Area_Chair_po3t · 2024-12-18

**Metareview:**

This paper presents a benchmark called DiffTell that is designed for the image difference captioning (IDC) task. The dataset consists of samples of a pair of images, each of which have been altered and the associated text caption describes the alteration. The authors train MLLMs on this dataset and show that the performance on the IDC task is improved.

Strengths
1. The paper proposes a clever trick to scale up collection of an IDC task by using alterations for both pairs of images.
2. Existing IDC datasets are limited in diversity and size. DiffTell improves on both these aspects.
3. The experiments in this paper show that on two datasets (IER, PSBattle), training MLLMs on DiffTell improves performance.

Weaknesses
1. The paper talks about IDC in the abstract (and other sections) but focusses only on the image manipulation aspect, and is tested only on those datasets. This makes the overall claim of the paper weaker and more general than supported by the experiments.
2. Since the data in DiffTell is synthetically generated, showing that training on it generalizes to real images is important. This is missing from the work. Given that DiffTell has short captions, this is very important.
3. Why do the authors need to combine DiffTell with the IER dataset for training? Does this point to some domain gap?


Justification of decision
Given the weaknesses of the work (narrow focus on IER not IDC, synthetic training data not shown to work on real datasets), I recommend the paper for rejection. The majority opinion of the reviewers is the same.

**Additional Comments On Reviewer Discussion:**

The reviewers raised concerns about the task and the dataset proposed in the paper as being too narrow and not evaluated thoroughly. The reviewers responded suggesting that they would include a more general IDC benchmark in a revised version of the paper. However, this was not supplied at the time of rebuttal.

---

### Decision · Program_Chairs · 2025-01-22

Reject